# Drivers of Stunting Reduction in Yogyakarta, Indonesia: A Case Study

**DOI:** 10.3390/ijerph192416497

**Published:** 2022-12-08

**Authors:** Tri Siswati, Slamet Iskandar, Nova Pramestuti, Jarohman Raharjo, Agus Kharmayana Rubaya, Bayu Satria Wiratama

**Affiliations:** 1Department of Nutrition, Poltekkes Kemenkes Yogyakarta, Tata Bumi No. 3, Banyuraden, Gamping, Sleman, Yogyakarta 55293, Indonesia; 2Center of Excellence for Applied Technology Innovation in the Field of Public Health, Poltekkes Kemenkes Yogyakarta, Tata Bumi No. 3 Banyuraden, Gamping, Sleman, Yogyakarta 55293, Indonesia; 3Balai Litbang Kesehatan Banjarnegara, Selamanik No. 16 A, Banjarnegara 53415, Indonesia; 4Department of Environmental Health, Poltekkes Kemenkes Yogyakarta, Tata Bumi No. 3, Banyuraden, Gamping, Sleman, Yogyakarta 55293, Indonesia; 5Department of Epidemiology, Biostatistics and Population Health, Faculty of Medicine, Public Health and Nursing, Universitas Gadjah Mada, Yogyakarta 55281, Indonesia; 6Graduate Institute of Injury Prevention and Control, College of Public Health, Taipei Medical University, Taipei 110, Taiwan

**Keywords:** HAZ, Indonesia, Yogyakarta, children, linear growth, stunting, intervention

## Abstract

Background: Chronic malnutrition in children is a severe global health concern. In Yogyakarta, the number of children who are too short for their age has dropped dramatically over the past few decades. Objective: To perform an analysis of trends, policies, and programs; and an assessment of government, community, household, and individual drivers of the stunting reduction in Yogyakarta, Indonesia. Method: Using a mixed-methods approach, there were three types of research: (1) analysis of quantitative data, (2) evaluation of stunting policy, and (3) focus group discussions and in-depth interviews to collect qualitative data. Results: The prevalence of stunting has decreased from year to year. Mean height-for-age z-scores (HAZ) improved by 0.22 SDs from 2013 to 2021. Male and female toddlers aged <20 months have relatively the same body length as the WHO median, but it is lower for children >20 months old. The COVID-19 pandemic has contributed to an increase in stunting-concurrent wasting. Nutrition-specific and -sensitive interventions have been carried out with coverage that continues to increase from year to year, although in 2020, or at the beginning of the COVID-19 pandemic, the coverage of specific interventions decreased. The government has committed to tackling stunting by implementing the five pillars of stunting prevention and the eight convergent stunting actions. As the drivers of stunting reduction, national and community stakeholders and mothers, at the village level, cited a combination of poverty reduction, years of formal education, prevention of early marriage, access to food, enhanced knowledge and perception, and increased access to sanitation and hygiene. Conclusions: Nutrition-specific and -sensitive sector improvements have been crucial for decreasing stunting in Yogyakarta, particularly in the areas of poverty reduction, food access, preventing child marriage, sanitation, education, and increasing knowledge and perception.

## 1. Introduction

Chronic malnutrition in early childhood causes stunting, which can damage children’s mental and physical development, and affect the intergenerational transmission of malnutrition and poor birth outcomes in the following generation. Stunting is an indicator of an inadequate birth and upbringing environment, and it is related to learning challenges and hurdles to community engagement. Due to this, the prevalence and severity of stunting is a useful indicator for population assessment and might be used to track the development of children in a population over time. In 2021, it was estimated that there were 149.2 million stunted children worldwide, or approximately 22 percent of all five-year-old children [1]. Specifically, 24.4% of children stunted in Indonesia [2].

Yogyakarta is the province which has the third-lowest stunting prevalence, after Bali and Jakarta. Based on the Indonesian Nutrition Status Survey in 2021, the prevalence of stunting in this province was 17.3%, but ranging from 14.1 to 20.6% throughout (Figure 1). The prevalence of stunting continues to decline from year to year. Between 2007 and 2018, the average decline was 0.57%, and between 2018 and 2021 it was 2.06%. The target provisioned according to the national middle-term development plan for 2020–2024, which refers to the SDGs for 2024, is 14% (Figure 2).

In an effort to achieve the acceleration of stunting reduction according to the specified target, the Indonesian government implemented the five pillars of the National Strategy for Stunting Reduction, which was established in 2018. The national policymakers and the World Bank have debated the Five Pillars Strategy, which was developed based on Indonesian knowledge and global best practices to combat stunting, including: political commitment and national leadership; a national communication campaign; a convergent national program with regional and community programs; nutrition and food security policy; and monitoring and evaluation. The implementation of convergence actions to reduce stunting is carried out through the eight convergence actions, namely: situation analysis (activity plan), action planning, discussion of stunting, regulation of village roles, human development, cadre development, a data management system, measurement and publication of stunting data, and annual performance reviews.

Many factors contribute to stunting children; e.g., the frequency of pregnant women with chronic energy malnutrition (CEM) increased from 10.7% in 2017 to 12.96% in 2020, conditions that have the potential to add the prevalence of low birth weight (LBW) increased from 5.2% in 2016 to 6.12% in 2020, the coverage of blood tablets for pregnant women dropped from 90.4% in 2016 to 87.9% in 2020, the antenatal care (ANC) decreased from 92.2 to 86.9% in 2020, childbirth in health facilities decreased from 99.8% in 2016 to 99.6% in 2022, and wasting toddlers increased from 4.05% in 2016 to 4.1% in 2020. The anemia rate among pregnant women also increased from 14.32% in 2017 to 15.84% in 2020. In fact, the occurrence of households with a risk of stunting is 77.1%; specifically, 57.35% (Sleman), 74.9% (Bantul), 77% (Kulon Progo), 80.6% (Gunung Kidul) and 82.9% (Yogyakarta City) [3].

On the other hand, some social factors that are related to stunting in toddlers are the following: the human development index (HDI) increased from 76.4% in 2013 to 80.2% in 2021; economic growth decreased from 5.05% in 2016 to −2.60% in 2020; the specific index of stunting handling increased from 78.5% in 2018 to 79.9% in 2019, the mean number of years of school increased from 8.51 in 2010 to 9.64 in 2021; people occupying decent housing increased from 86.9% in 2017 to 92.3% in 2020; and the number of clean and healthy living people increased from 42% in 2020 to 53.5% in 2021 [3].

The Director of the WHO in March 2020 stated that COVID-19, caused by SARS-CoV-2, spread throughout the world and announced the conditions of the global pandemic outbreak [4]. Meanwhile, the President of Indonesia announced the first COVID-19 case in Indonesia on 2 March 2020. This pandemic has had a far-reaching impact on all human lives, including the impact on stunting-reduction indicators. The most severe indicators of impact, from major to minor, are growth monitoring, the vitamin A program, extra feeding programs, and child and baby feeding programs. Some of the quantitative data during January 2020 to September 2020 show that coverage of supplementary feeding for children dropped from 3.4 to 0.3%; coverage of children who increased their weight (N/D) dropped from 4.6 to 0.3; and stunting, underweight, and wasting prevalences decreased from 27.6 to 13.1%, 20.6 to 7.5%, and 12.6 to 4.1%, respectively [5].

Some studies conducted in Yogyakarta have proved the risk factors of stunting comprises of maternal body height [6], maternal knowledge of feeding [7,8], a pregnant mother having anemia, a history of low birth weight, food insecurity, a stunted mother [8], low birth weight [7], breast feeding and exclusive breast feeding [9,10,11], complementary food to breast feeding [12,13], and family SES [14]. There are many surveys on why stunting is getting less common in Yogyakarta, and they refer to a few key factors of both specific and sensitive interventions: parental education [7]; health care interventions, such as delivery by a skilled birth attendant (SBA), antenatal care, and postnatal care; improvements in water, sanitation, and hygiene (WASH) through clean and healthy living behaviors, such as accessing better water sources.

The goal of this study was to do a full analysis of the factors at the national, community, household, and individual levels that led to a drop in stunting in Yogyakarta.

## 2. Methods

### 2.1. Study Design

This was a mixed-method approach with three studies, including a quantitative data analysis, a comprehensive review of nutrition-specific and -sensitive policies and programs, and a qualitative data collection and analysis process.

### 2.2. Quantitative Methods

#### 2.2.1. Data Sources

The sources of data were from the Basic Health Survey in 2013 and 2108 for representing the condition before the COVID-19 pandemic, and the Indonesian Nutritional Status Survey in 2021 for representing the condition during the COVID-19 pandemic.

#### 2.2.2. Outcomes

A quantitative method was used to determine the trend of stunting prevalence, stunting and its coexistence, and the height-for-age z score (HAZ) before and during the COVID-19 pandemic compared to WHO’s median.

#### 2.2.3. Setting

This analysis covers all districts in Yogyakarta Province.

#### 2.2.4. Statistical Analysis

Children’s HAZ kernel density plots were calculated for all survey years in order to study the demographic variations in slowing growth through time. Standardized and well-established methodologies for conducting equity analyses were utilized to examine the prevalence of stunting by mother’s education and place of living (urban vs. rural).

### 2.3. Qualitative Method

#### 2.3.1. Data Source

The qualitative method was used to describe policies and programs through desk review; to describe stunting prevention programs, for both sensitive and specific interventions; and to investigate how the pandemic affects stunting reduction efforts, convergence of stunting, and best practices in successfully tackling stunting and its obstacles through focus-group discussions and in-depth interviews with stakeholder experts at the provincial, district, and sub-district levels.

#### 2.3.2. Setting

Both the FGDs and in-depth interviews were conducted in districts with the highest and lowest prevalences of stunting, and districts and villages with the highest and lowest proportions of locus and non-locus stunting.

#### 2.3.3. Analysis

These frameworks led to the qualitative study and interpretation of the important determinants, contextual factors, barriers, and facilitators of nutrition-specific and -sensitive events and convergent stunting. Using the method of thematic analysis, important themes that emerged regarding the causes of stunting were investigated.

#### 2.3.4. Ethical Consideration

This investigation was approved by the IRB Poltekkes Kemenkes Yogyakarta Ministry of Health Yogyakarta: e-KEPK/POLKESYO/0223/II/2022, dated 23 February 2022.

## 3. Results

### 3.1. Characteristic of Sample

Most of the children in the 2013, 2018, and 2021 surveys were aged 24–59 months, female, and lived in urban areas. Details are in Table 1.

### 3.2. HAZ Kernel Density Plots, HAZ Scores, and Prevalence of Stunting

Based on Figure 3, the description is that the rightward shifting and curve narrowing over time point to nutritional benefits for all children under the age under 5 y during the 8 y period of study. The greatest difference occurred between 2018 and 2021; the other changes were more modest increases in HAZ.

The average HAZ score among 0–59 month children from 2013 to 2021 increased by 0.22 SDs, from −0.93 SDs to −0.71 SDs. In contrast, the increase in HAZ score in stunted children was very slow, by −0.19 SDs. The HAZ score for stunted children in 2021 was still quite severe at −2.55 SDs, far from the normal category limit of −2 SDs in 2021 (Figure 4).

The average body length or height of male and female toddlers in 2021 increased compared to the previous study. Male and female toddlers aged <20 months had relatively the same body length as the WHO median. However, after >20 months, the heights of the male and female toddlers were lower than the WHO median. The height of the males was still higher than that of the females (Figure 5A,B).

Although there was a decrease in stunting, the COVID-19 pandemic contributed to acute malnutrition in the form of stunting-concurrent wasting (Figure 6).

### 3.3. Program Intervention

The program for tackling stunting in children includes sensitive and specific interventions. Sensitive interventions address indirect stunting causes outside health issues, including improvement of drinking water and sanitation facilities, nutrition and health services, nutrition knowledge, and nutritious food access. Specific interventions are activities that directly address the causes of stunting and are generally provided by the health sector, such as food intake, infection prevention, maternal nutritional status, infectious disease treatment, and environmental health. In fact, the coverage of intervention programs, both sensitive and specific, increased year by year from 2018 to 2021 (Figure 7A,B).

### 3.4. The Qualitative Results

#### 3.4.1. Overview of Anti-Stunting Policies

As a member state of the United Nations, Indonesia has committed to reduce the prevalence of stunting, which a public health concern. The Presidential Regulation Number 72 of 2021, Acceleration of Stunting Reduction in Indonesia, is expected to further accelerate the reduction of stunting, as one of the strategies in Scaling Up Nutrition (SUN). The National Strategy for Stunting Acceleration is carried out through five pillars and eight convergence actions to overcome stunting through sensitive and specific interventions. We provide an overview of the legislation, policies, initiatives, and enablers in Yogyakarta Province and Indonesia from 2021 to 2024 (Figure 8).

As support for national regulations and the commitment of local governments at the district/city level, there are several programs which encourage efforts to accelerate stunting reduction, including the local action plans for food and nutrition; community movements for healthy living; exclusive breastfeeding and breastfeeding; non-smoking areas; KRPL (Kawasan Rumah Pangan Lestari or in English: Food Household Area Program); a diet that uses a food arrangement for one meal or for a day according to mealtimes (morning, afternoon, and evening/night) which contains nutrients to meet the needs of the body in amounts that meet the rules of balanced nutrition in accordance with the acceptability (taste, culture) and purchasing power of the community and is safe for consumption or in Indonesia—namely, Bersih, Beragam, Seimbang, and Aman (B2SA or in English: awareness of various nutritious, balanced and safe food movements); child-friendly areas; education of brides-to-be through the Elsimil application; the Dahsat program (Dapur Sehat or in English: Healthy Kitchen); the e-HDW (electronic human development worker) application; and other innovations based on local wisdom.

Currently, all areas in Yogyakarta Province have become loci of anti-stunting efforts to encourage a greater reduction in stunting. In addition, a multisectoral approach, strong advocacy, and political leadership support the reduction of stunting. The role of the penta-helix in reducing stunting is very necessary. Some of the optimization options are cooperation with NGOs and CSR in terms of program funding; cooperation with universities and professional organizations for determining the programs for activities and curricula; the implementation of the three pillars of higher education in the Inter Professional Collaboration (IPC), Inter Professional Education (IPE), and policy dissemination platforms or forums, social media, and the government.

Although stunting has decreased, there are still several obstacles to reducing stunting. At the program level, the obstacles encountered are the difficulty of coordination, strategies that are not strong enough, lack of interest from stakeholders, unequal collaboration structures, limited human resources, and budget availability. Meanwhile, at the household and individual levels, the problems are related to common opinions related to the causes, obstacles, and challenges of stunting in Yogyakarta, including maternal education, maternal knowledge and practices in providing parenting and food diversity to toddlers, the economy, poverty, food security at the household level, and the prevention of early marriage.

#### 3.4.2. Qualitative Inquiry Results from Expert Stakeholders

By 2018, all districts had implemented a comprehensive nutrition intervention policy addressing maternal, infant, and young-child nutrition. However, coordinating the many sectors and actors involved in treating malnutrition is difficult, so district-level multisectoral coordination systems are needed. All of the local organizations (in Indonesia, namely, Organisasi Perangkat Daerah or OPD) understand the necessity of a multisectoral approach to tackle malnutrition. Health, agriculture, education, family planning, water, and sanitation play major roles in national nutrition policies and plans.

Health ministries are involved in all multisectoral coordination mechanisms throughout the area in most cases. All intervention nutrition programs must be integrated into healthcare, as universal access to prenatal and postnatal care is crucial. Strong, accessible health systems are important to administering nutrition treatments; maternal health, starting with adolescent females; and promoting infant young-child feeding (IYCF).

Some acknowledge education’s influence on children’s health care and wellbeing. Several causes of stunting can be addressed through education, including by increased childcare knowledge, practice with feeding, health seeking behavior, and the opportunity to earn a higher income.

Agriculture is a part of the coordinating systems that are used for nutrition interventions. However, having access to a variety of micronutrient-rich foods, such as zinc-enriched rice, fish, vegetable seeds, and omega-3-enriched eggs, and education on how to prepare food, is critical for preventing stunting in children. There is a significant amount of effort that needs to be done in order to sustainably bring variety to the food supply. A strategy for achieving this goal is to enhance the food security of households in order to facilitate the preparation of foods containing greater concentrations of nutrients.

Sanitation facilities, such as waste management one and toilets, and the availability of clean water and drinking water, are important aspects supporting infection prevention to achieve optimal child health. In terms of reaching the target of 1000 early days of life, there are families who do not have their own toilets, and the goal of making Indonesia open defecation-free (ODF) is not met. In addition, the waste problem needs to be tackled by the people using community engagement methods. Waste that is not managed effectively is a cause of disease, whereas when it is managed well, it will generate economic benefits, a good environment, and prevent diseases.

The National Population and Family Planning Agency is the institution that has the task of controlling the population through the implementation of population- and family-planning programs, and improving the quality of Indonesia’s human resources through family development. Since 2021, BKKBN has been appointed as the coordinating chair of the coordinator for the acceleration of stunting reduction based on Presidential Regulation Number 72 of 2021. The innovation in family-based stunting prevention is education through the Family Development for Toddlers to Eliminate the Problem of Stunting Children (in Indonesia, namely, Bina Keluarga Balita Eliminasi Masalah Anak Stunting or abbreviated BKB EMAS).

Expert stakeholders pointed out that to reduce stunting problems, it is important to pay attention to food insecurity, improvements in education, women’s empowerment, increases in remittances, improvements in the standard of living and nutrition, poverty reduction, and the availability of sanitation. During interviews with stakeholders, nutrition-specific and nutrition-sensitive policies and initiatives were considered important. Overall, the necessity of a diverse mix of measures that target both specific outcomes, such as maternal health, newborn health, and nutrition; and broader efforts, such as poverty reduction and education, were emphasized. Among the underlying causes of stunting reduction, expert stakeholders focused on improving nutrition, primarily in terms of food diversity, parenting, and maternal knowledge and perceptions of stunting.

All regional organizations have made stunting a top priority, and all regional heads have made a strong commitment to reducing stunting. However, there is still a lack of coordination and different ideas about who is responsible for stunting programs, especially in the non-health sector. As stated below:


*“All local government organizations have committed to reducing stunting and have carried out 8 convergent stunting actions, but there are still some weaknesses, such as coordination and misperceptions, especially in the non-health sector in efforts to reduce stunting”.*


In contrast, in Sleman District, which has a low prevalence of stunting, coordination and cooperation for stunting prevention are conducted with the involvement of social services, health offices, and traditional markets to provide extra feeding for toddlers.

#### 3.4.3. Communities Stake Holders

We compiled data from community stakeholders, including doctors, Puskesmas (community health center) directors, district, and sub-district administrators. They described several causes of stunting related to family economic status, maternal parenting practices, food diversity, smoking, and early marriage.

Furthermore, related to specific interventions at the Puskesmas level, we innovated a more detailed examination of stunting toddlers with more intense health monitoring by pediatricians and examinations of pregnant women by obstetricians. We also have innovations to overcome stunting by involving adolescents and pregnant women in their educational efforts to delay the age of marriage.

We also suggested latrines in every household of the first 1000 days of life. It was reported that if they did not have a toilet or still used a toilet together with other households, the locals worked with the local General Works Office to make a latrine. However, the obstacle that exists is precisely the behavior of people, mainly in rural areas, who prefer to defecate in rivers over toilets.

For underlying causes, many community health workers complain about smoking among fathers. They do not know or do not want to know that smoking is related to stunting toddlers. In addition, the issue of the increase in cigarette prices is not an obstacle for smokers. Although there are regulations and declarations of non-smoking areas, they are not accompanied by strict sanctions, so that there is no social responsibility for people who smoke near toddlers or pregnant women. As stated by a public health official:


*“Smoking is still an issue that interferes with stunting reduction. Whether it is related to the proportion of food spending for toddlers or aspects of its impact on vulnerable groups, sometimes they still find them smoking freely in the house, smoking while holding their children or smoking near pregnant women”.*



*“In addition to these factors, we are very constrained in overcoming community-based stunting because mothers do not understand stunting; they deny if their children are called stunted, especially if they belong to a family that is economically good”.*


#### 3.4.4. Mothers in Communities

Among the contextual variables highlighted by maternal children were rises in social economic knowledge, children’s health care, and poverty reduction strategies to combat malnutrition. While the underlying causes of the decline in stunting were unknown, mothers cited increases in food security through extra feeding during pregnancy and infancy, and access to health care during pandemics. As stated,


*“This pandemic has aggravated the economic situation of the community, especially for informal workers. The decline in income has an impact on food spending, especially on toddler food. For this reason, I really hope that the provision of food assistance from the government will enable me to make a living. We need extra feeding for toddlers and pregnant women so as to prevent stunting”.*


## 4. Discussion

Stunting in Yogyakarta Province has decreased, both in terms of the prevalence and by an improvement in HAZ score. In the period 2013–2018, the prevalence of stunting in DIY fell from 27.2% to 21.42%, or decreased by an average of 0.57% per year. In 2018–2021, the prevalence fell from 21.42% to 17.3%, or an average decrease of 2.06%—an almost 4-times-higher rate. Stunting decreased by 10% from 2003 to 2021. If the speed of stunting reduction in Yogyakarta in the next three years is consistent, it is estimated that by 2024 the prevalence of stunting will be 11.12%, so that the target of reducing stunting by 14% will be exceeded [15]. It is even estimated that Yogyakarta will be independent of stunting before 2030.

Based on the HAZ score, the severity of stunting children and all children under five has been reduced, as depicted by the improvement in the HAZ score (Figure 3). The improvement in the HAZ score occurred after the government implemented a national strategy for stunting prevention through five pillars and eight convergent stunting actions in 2018 [16]. Several other provinces have had similar experiences in terms of the impacts of convergence actions on the reduction of stunting in toddlers, such as in South Sulawesi [17] and Central Java [18]. In 2011, the Indonesian government made a strong commitment to stunting mitigation by joining the global Scaling Up Nutrition (SUN) movement—a global effort from some countries to strengthen the commitments and the action plans for scaling up nutrition [19]. Then, it was strengthened by releasing the national strategy to accelerate stunting reduction and prevention using a multi-sectoral approach through a convergence program at all levels [20]. In addition, in 2018, the Indonesian government released the national strategy to accelerate stunting reduction [16]. Convergent actions taken to hasten the elimination of stunting will produce better results than parallel actions taken in either direction [21].

The experience of several countries that had already been successful in reducing the prevalence of stunting was based on a strong commitment from the government [22] in arranging policies and their implementations of sustainable political commitments, multi-sectoral approaches, organizational regulation at all levels, and improved access to good-quality health services [23]. The integration of sensitive and specific interventions has the potential to lower the prevalence of stunting by between 0.7 and 2.1% each year throughout the world [21,24]. In fact, after the stunting convergence action was implemented, the acceleration of stunting reduction in Yogyakarta reached 2.06% per year during 2018–2021, four times faster than the previous year (2007–2018)—that is, by 0.5% per year. Several countries experienced accelerated reduction in stunting at different speeds. For example, Maharasta, India, was down by 2.6% [25].

Although there has been a decline in the HAZ score from year to year, the decline still seems slow. The improvement in HAZ score among toddlers from 2013 to 2021 was only −0.19 SD. The low of HAZ score in stunting toddlers is related to the retained effect of chronic malnutrition suffered by mothers during pregnancy and the success of nutritional interventions in the first 1000 days of life [26]. The potential impacts of the lasting effect are intergenerational stunting and intergenerational poverty, indicating that there are still social problems, such as poverty, inequity, and inequality [27,28,29].

The study also reported that the prevalence of stunting-concurrent wasting children increasing during the COVID-19 pandemic. It can be explained by the COVID-19 pandemic (a) decreasing the rate of economic growth; (b) increasing poverty; (c) shifting working status from formal to informal; (d) causing the instability of growth monitoring and healthcare through Posyandu services during the adaptation to the pandemic; (e) and reducing the coverage of complementary feeding.

Nutrition is being scaled up through the efforts of the National Movement for the first 1000 days of life by involving cross-sectoral partners [30,31,32]. The Yogyakarta government puts stunting as a priority issue. The government unites all steps across programs and sectors through a national strategy of the five pillars and eight convergence actions. Regulations at the provincial and district levels for preventing and reducing stunting in Yogyakarta include: (a) acceleration of stunting reduction through specific and sensitive interventions; (b) providing lactation rooms in workplaces and public places. Stunting prevention starts with efforts to fulfill nutrients in the first 1000 days of life. Exclusive breastfeeding provides irreplaceable essential nutrients for the growth and development of children. All of these issues are described in the stunting prevention action plan established by Governor of DIY No. 92 of 2020 [33].

In general, the prevalence of stunting in Sleman Regency in 2020 was the lowest among the districts in Yogyakarta [17], and all sub-districts have reached a prevalence of <14%. Several social factors explain the low prevalence of stunting in Sleman Regency related to the higher well-being indicator, including: (1) the human development index (HDI), (2) years of education, (3) economic growth, (4) coverage of maternal and pregnant women’s healthcare, (5) coverage of neonatal healthcare, (6) coverage of exclusive breast feeding, (7) the low cases of LBW, and (8) innovations to reduce stunting—i.e., GeTAR Thala (Gerakan Tanggulangi Anemia dan Thalasemia Remaja, English: Adolesence Anemia and Thalasemia Tackling Movement), PANdu TEMan (Pelayanan Antenatal Care Terpadu Menuju Triple Eliminasi Melibatkan Semua Layanan, English: Integrated Antenatal Care for Triple Elimination Involving All Services), Pecah Ranting Hiburane Rakyat (Pencegahan Rawan Stunting Hilangkan Gizi Buruk Tingkatkan Ekonomi Rakyat, English: Preventing Stunting Improve Economic Community) and Gambang Stunting (Gerakan Ajak Menimbang Cegah dan Atasi Stunting, English: Weight Movement Prevent and Address Stunting) [34]. Meanwhile, the high prevalence of stunting in Bantul District, especially in Sub-District Dlingo, is related to the high frequency of early marriage, low level of education, economic issues, food diversity, improper feeding practices, and relatively lower incomes than other areas in Yogyakarta Province. All levels of government have committed to tackling stunting through the allocation of village funds. Some (a small option) of OPDs, especially at the regional level, still consider stunting a serious health problem.

## 5. Conclusions

In conclusion, the situation in Yogyakarta is representative of a province that was able to rapidly reduce the under-5 stunting prevalence after a period of stagnation. This is attributable to a variety of aspects, including effective leadership, a strong civil society, and the targeting of vulnerable communities during the implementation of specific and sensitive nutrition interventions. The capacity to govern in a favorable environment is characterized by the impetus of programmatic initiatives, including effective and sustained political leadership, active participation of civil society in the design and implementation of policies and programs, an emphasis on adequate accountability mechanisms at all levels, and sustained and equitable implementation of out-of-sector and within-sector evidence-based interventions. Although stunting has decreased, there is still a need for better coordination through multi-sector, multi-actor, and strategic collaboration to improve human resources and budgets, and improve parenting, food diversity, household food security, and early marriage prevention.

## Figures and Tables

**Figure 1 ijerph-19-16497-f001:**
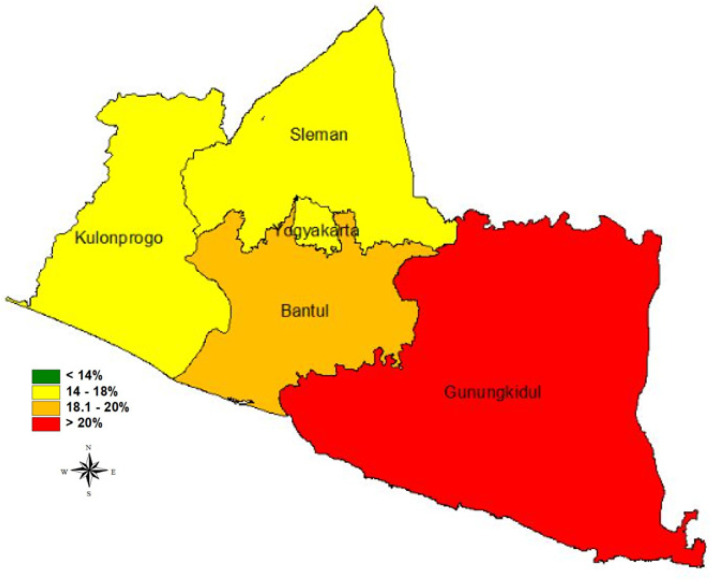
Map of stunted children under five in Yogyakarta Province, 2021.

**Figure 2 ijerph-19-16497-f002:**
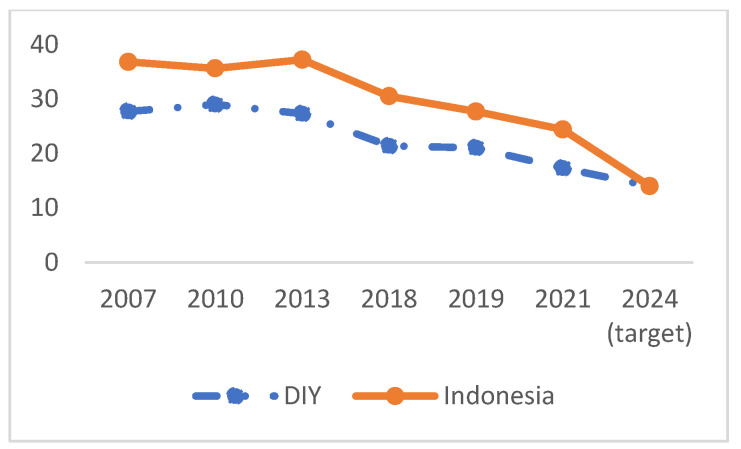
Trends of under five stunting prevalence in Yogyakarta Province (2007–2021).

**Figure 3 ijerph-19-16497-f003:**
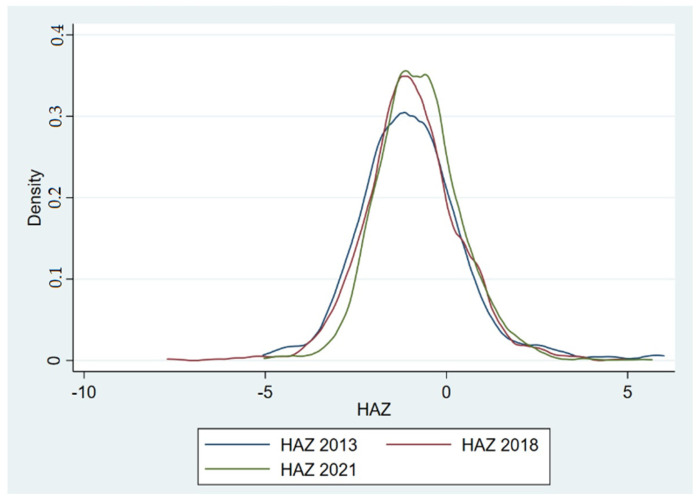
HAZ kernel density plots of 0–59 month children during 2013–2021.

**Figure 4 ijerph-19-16497-f004:**
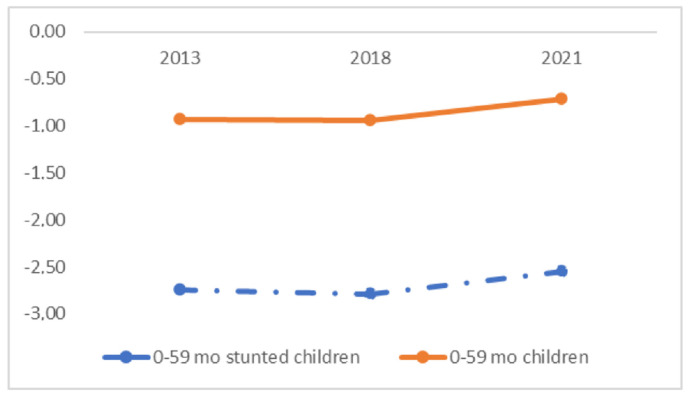
Trend of HAZ score among children and stunted children in Yogyakarta, 2013–2021.

**Figure 5 ijerph-19-16497-f005:**
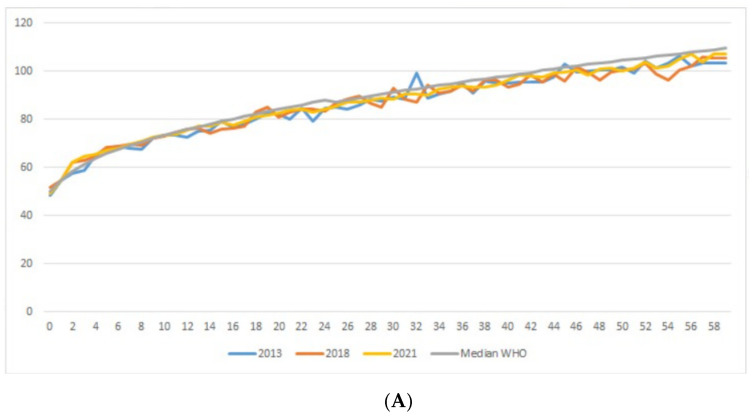
(**A**) Trend of average length/height of male toddlers in Yogyakarta, 2013–2021. (**B**) Trend of average length/height of female toddlers in Yogyakarta, 2013–2021.

**Figure 6 ijerph-19-16497-f006:**
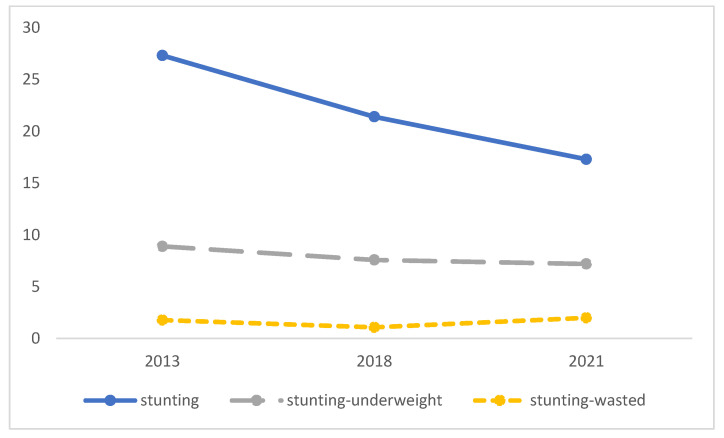
Trend of stunting and its concurrence in Yogyakarta, 2013–2021.

**Figure 7 ijerph-19-16497-f007:**
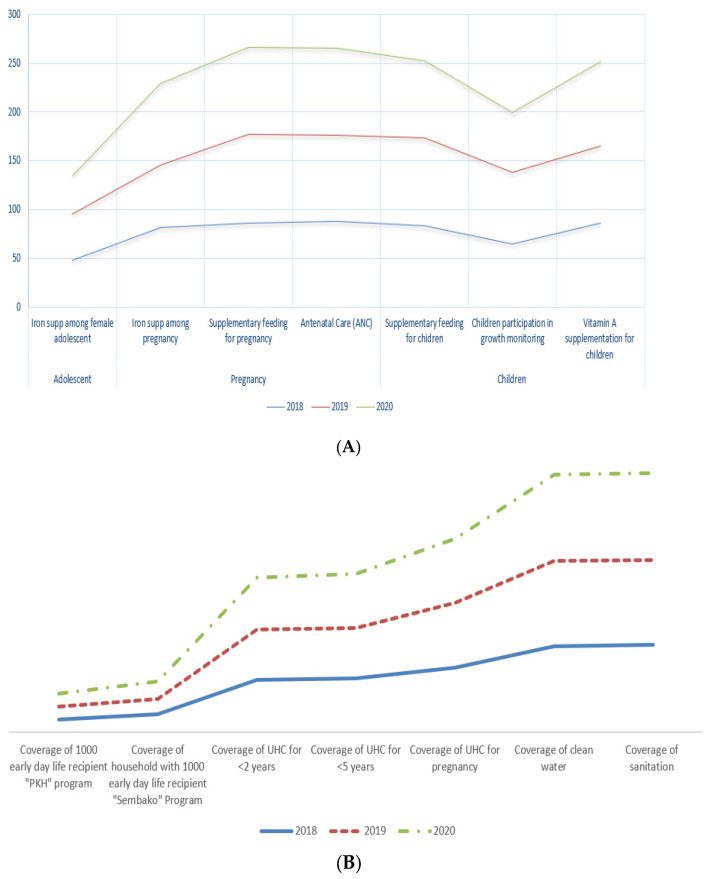
(**A**) Coverage of specific interventions for tackling stunting in children in Yogyakarta, 2018–2020; source: bangda.go.id. (**B**) Coverage of sensitive interventions for tackling stunting in children in Yogyakarta, 2018–2020; source: bangda.go.id.

**Figure 8 ijerph-19-16497-f008:**
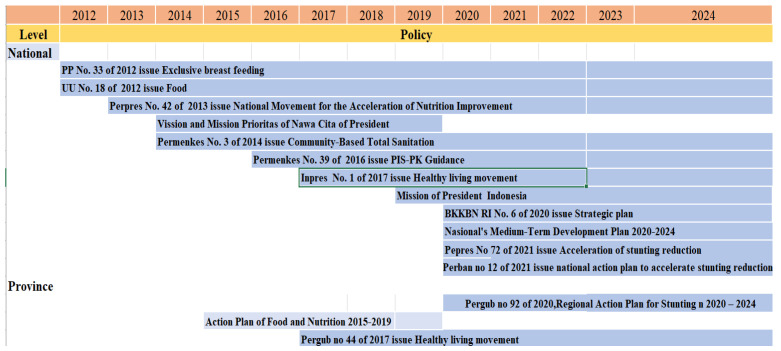
Overview of legislation, policies, initiatives, and enablers 2012 and 2024 in Yogyakarta Province and Indonesia.

**Table 1 ijerph-19-16497-t001:** Sample characteristics.

Characteristics	2013 (*n* = 795)		2018 (*n* = 731)		2021 (*n* = 2877)		Total	
	*n*	%	*n*	%	*n*	%	*n*	%
Age (mo)								
<6	54	7.7	60	8.2	168	5.8	282	6.5
6–23	201	28.5	210	28.7	876	30.5	1287	29.8
24–59	450	63.8	461	63.1	1833	63.7	2744	63.6
District								
Kulon Progo	174	24.7	142	11.5	450	11.2	592	11.3
Bantul	125	17.7	164	24.5	737	28.4	901	27.5
Gunung Kidul	125	17.7	149	18.1	578	17.5	727	17.8
Sleman	155	22.0	189	33.6	692	32.9	881	33.3
Yogyakarta City	126	17.9	87	10.2	420	9.9	507	10.1
Gender								
Male	377	53.5	385	51.8	1463	50.6	1848	51.2
Female	328	46.5	346	48.17	1414	49.4	1760	48.8
Setting								
Urban	529	75.1	497	74.8	1997	75.7	2494	75.21
Rural	176	24.9	234	25.3	880	24.4	1114	24.8

## Data Availability

All data and models of study are available from the corresponding author upon reasonable request.

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
