# Peer review of "Drivers of Stunting Reduction in Yogyakarta, Indonesia: A Case Study"

_ijerph, 2022, doi:10.3390/ijerph192416497_

Round 1
Reviewer 1 Report
Hope this study will contribute to the existing literature in many aspects but I have certain concerns which are listed below:
Line 24 it is Result or Results?
Line 27 what is stunted wasted? Correct it. Do you want to say stunting and wasting? Also, see line 409 stunting wasted?
Line 82 authors wrote ‘Those factors lead to stunted children.’ which factors? Write clearly.
Line 86 sentence ended with a colon instead full stop. The next sentence on line 87 starts with a capital letter.
Line 397, 408, 416 Why every citation is in a separate bracket? Correct it as [20,23], [26-28], [29-31]. Also see lines 106 and 108
Libe 416 HPK? Write a full name for HPK. Line 422 HPK is used again here you can use an abbreviation.
Line 429 HDI is used. Maybe it is the human development index. On line, 87 authors used the human development index but did not put HDI in brackets here. Give the abbreviation HDI very next to the term human development index at line 87.
Line 238-243 correct run-on sentences. Write clearly and shorten sentences.
The conclusion section needs to be improved
Author Response
Reviewer 1
Comments and Suggestions for Authors
Hope this study will contribute to the existing literature in many aspects but I have certain concerns which are listed below:
Line 24 it is Result or Results?
- Author respond: We appreciate the reviewer comments, we replace results
Line 27 what is stunted wasted? Correct it. Do you want to say stunting and wasting? Also, see line 409 stunting wasted?
- Author respond:
We appreciate the reviewer comments:
Line 27:stunting-wasting
Line 409 : stunting wasting
Line 82 authors wrote ‘Those factors lead to stunted children.’ which factors? Write clearly.
- Author respond:
We appreciate the reviewer comments: We improved the sentences
Line 86 sentence ended with a colon instead full stop. The next sentence on line 87 starts with a capital letter.
- Author respond:
We appreciate the reviewer comments: we improved using lowercase
Line 397, 408, 416 Why every citation is in a separate bracket? Correct it as [20,23], [26-28], [29-31]. Also see lines 106 and 108
- Author respond:
We appreciate the reviewer comments: we improvement them.
Line 416 HPK? Write a full name for HPK. Line 422 HPK is used again here you can use an abbreviation.
- Author respond:
We appreciate the reviewer comments: we improved by the first 1000 days of life
Line 429 HDI is used. Maybe it is the human development index. On line, 87 authors used the human development index but did not put HDI in brackets here. Give the abbreviation HDI very next to the term human development index at line 87.
- Author respond:
We appreciate the reviewer comments: we improved
Line 238-243 correct run-on sentences. Write clearly and shorten sentences.
- Author respond:
We appreciate the reviewer comments: we improved as below:
Some activities that have potentiality to be optimized are cooperation with NGOs and CSR in terms of program funding, as well as collaboration between and among universities and professional organizations in determining the program of activities and curriculum, the implementation of the three pillars of higher education in the form of Inter Professional Collaboration (IPC), Inter Professional Education (IPE), as well as policy dissemination platforms through forums, social media use, and government channels.

Reviewer 2 Report
Well written paper, however needs language corrections and improvement.
Author Response
Reviewer: Well written paper, however needs language corrections and improvement.
Author respond: |
We appreciate the reviewer comments. Thank you for suggestion: We make some language correction and improvement

Reviewer 3 Report
Dear authors,
First of all, congratulations for your extensive research !
The mixed methods approach you have chosen is very challenging and must be valued more in the discussion, where the relation between the level of education of the mother (respectively, father’s education) and child future developmental problems, can be extensively presented. This study can be also improved by extending the conclusions section.
Author Response
We appreciate for this comment, we make improvement:
In conclusion, the Yogyakarta case is representative of a province that was able to rapidly reduce the under-5 stunting prevalence after a period of stagnation. This is at-tributable to a variety of aspects, including effective leadership, a strong civil society, and the targeting of vulnerable communities during the implementation of specific and sensitive nutrition interventions. The capacity to govern in a favorable environment is characterized by the impetus of programmatic initiatives, including effective and sus-tained political leadership, the active participation of civil society in the design and implementation of policies and programs, the emphasis on adequate accountability mechanisms at all levels, as well as the sustained and equitable implementation of out-of-sector and within-sector evidence-based interventions. Although stunting has decreased, there is still a need for better coordination among multi-sector, multi-actor, and strategy collaboration to improve human resources and budget, as well as boost excellent parenting and food diversity, household food security, and early marriage prevention.
